# An interpretative phenomenological study about maternal perceptions of cesarean birth

Hala Bawadi[1], Zaid Al-Hamdan[iD][2]*, Nagham Abu Shaqra[3], Amer Gharaibeh[4], Hasan Rawashdeh[iD][5], Asma Basha[6], Majida Jallad[7], Sawsan Majali[8], Heba AboShindi[9], Mahmoud Taym[iD][8]

**1** Department of Maternal and Child Health, School of Nursing, The University of Jordan, Amman, Jordan, **2** Department of Community and Mental Health, Faculty of Nursing, Jordan University of Science and Technology, Irbid, Jordan, **3** USAID Jordan, Amman, Jordan, **4** Department of Obstetrician and Gynecologist, Royal Medical Services, Amman, Jordan, **5** Jordan University of Science and Technology, Amman, Jordan, **6** Department of Obstetrics and Gynecology, The University of Jordan and Jordan University Hospital, Amman, Jordan, **7** Private Clinic, Istishari Hospital, Amman, Jordan, **8** USAID Health Services Quality Accelerator Activity, University Research Company, Amman, Jordan, **9** Data Management and MMSR Advisor, USAID Health Services Quality Accelerator Activity, Amman, Jordan

* zaid_hamdan@hotmail.com, zmhamdan@just.edu.jo

## Abstract

### Aim

To explore the perceptions of Jordanian primipara and para 1 women regarding Cesarean section delivery and provide insights for clinical practice, policy development, and future research to optimize maternity care practices and outcomes in Jordan and similar contexts.

### Background

Global Cesarean section (CS) rates have risen steadily over recent decades. Jordan has witnessed a significant increase in CS rates. This trend has sparked interest in understanding the underlying factors behind this phenomenon and the perceptions surrounding CS delivery. Understanding these factors is essential for informing healthcare practices and promoting optimal maternal and neonatal health outcomes.

### Methods

Forty-one primipara and para 1 women were interviewed within seven focus groups. The analysis was conducted using interpretative phenomenological analysis with the assistance of the NVivo 12 software program.

### Results

Three main themes were identified: The Foundation of Women's Knowledge and Cognitive Structures; The Influential Parties in Shaping Women's Perceptions; and Transforming Perception into Action.

**Data availability statement:** All relevant data are within the manuscript.

**Funding:** The author(s) received no specific funding for this work.

**Competing interests:** The authors have declared that no competing interests exist.

## Discussion

To offer empathetic, comprehensive, and patient-centered care, healthcare professionals should discuss delivery options with pregnant mothers and factor in maternal preferences and perceptions. This study underscores the importance of supportive dialogue and informed decision-making in maternity care, advocating for a more individualized approach to childbirth.

## Introduction

In Jordan, maternity care encompasses both public and private healthcare services, supporting women through pregnancy, childbirth, and the postpartum period. While many women utilize public health facilities due to their accessibility and affordability, a significant number choose private healthcare for its perceived higher quality and personalized care, despite the associated higher costs. Public hospitals often remain the primary choice for complex pregnancies or labor due to their comprehensive services [1].

Maternity care services are provided by obstetricians, nurses, and midwives across both healthcare sectors. Notably, 96% of women prefer to receive antenatal care from a doctor, with only 3% selecting for midwives or nurses, indicating a strong preference for physician-led services. Antenatal visits typically include essential check-ups, screenings, nutritional advice, and health education. The experience can vary based on the chosen healthcare setting: private facilities often provide longer consultations, which many women find more reassuring, while public services are favored for their affordability and availability [2].

The global rise in Cesarean Section (CS) rates has become a concern to healthcare providers, policymakers, and expectant mothers alike. The World Health Organization (WHO) states that an optimal CS rate falls between 10% and 15% of all births [3]. In Jordan, recent years have seen a rate of approximately 25% [4]. Data collected from different sectors of Jordan's healthcare system reported the following rates for 2015–2017: 40.4% (±2.6) in university hospitals, 39.1% (±1.8) in private hospitals, 36.1% (±0.2) in military hospitals, and 27.4% (±0.7) in governmental hospitals [5]. These statistics have prompted researchers to investigate the factors contributing to the prevalence of CS deliveries and to understand the perceptions of CS among pregnant Jordanian women, specifically those who are experiencing their first birth (primipara) or have previously given birth once (para 1).

Numerous factors influence a woman's choice to undergo a CS delivery. These factors include medical indications, cultural norms, socioeconomic status, and individual preferences. Salem (2021) pointed to previous CS (33.6%), abnormal presentation (20.3%), and patient request (16%) as common indications for CS in Jordan. CS, when necessary, is undoubtedly a life-saving intervention. However, when performed without medical justification, it can put both mother and baby at risk, leading to increased rates of maternal morbidity and mortality, neonatal respiratory complications, and longer recovery times [6]. Therefore, we need to understand how Jordanian primipara and para 1 women perceive CS delivery to promote birthing practices and achieve optimal maternal and neonatal outcomes.

Women's perceptions and decisions about CS are shaped by personal beliefs, cultural influences, and healthcare setting. Some women view CS as a necessary medical intervention to ensure the safety of both mother and baby, perceiving it as a modern advancement that offers controlled and planned delivery while potentially reducing the risks associated with normal vaginal delivery (NVD). However, this perception is often influenced by factors such as trust in doctors, advice from healthcare providers, and familial or cultural expectations, particularly in private hospitals where personalized care is emphasized [7].

Previous research on CS delivery has focused on healthcare professionals' perspectives rather than those of pregnant women. This qualitative study seeks to explore this population's perceptions, beliefs, and decision-making processes surrounding CS. This will enable healthcare providers to tailor care to serve expectant mothers better, ultimately promoting evidence-based, woman-centered maternity care in Jordan.

This qualitative study investigated the perceptions of Jordanian primipara and para 1 women regarding CS delivery. Focus group interviews were conducted to reveal participants' attitudes and knowledge about CS, its risks and benefits, and their reasons for opting for or rejecting this mode of delivery. The findings serve to inform clinical practice, policy development, and future research endeavors to enhance maternity care practices and outcomes in Jordan and similar contexts.

## Methods

### Study design

An interpretative phenomenological approach was adopted because it allows participants to respond to in-depth, probing questions, express their opinions, and expand upon their experiences [8].

### Setting and participants

This Jordanian study recruited its participants from two public hospitals and two health centers. Three focus groups were held with primiparous women (2nd and third trimesters) at hospital antenatal care (ANC) clinics. Four other focus groups were conducted with para 1 women (14–50 days postpartum) who brought their infants to the centers for vaccination and neonatal screening. Purposive sampling was employed, utilizing a maximum variation strategy [9]. Based on age, education level, and family income. In total, 41 women who met the criteria and agreed to participate were divided into seven groups of 5–7, the sample size was determined based on data saturation [10].

### Data collection and procedures

Focus groups interviews were used for data collection as they are effective approaches for exploring perceptions and lived experiences [11]. Participants expressed themselves freely, and the researchers posed questions uncovering in-depth data. To ensure comprehensive coverage of relevant topics [12]. The research team developed an interview guide, as shown in Table 1. The lead author conducted the interviews, each lasting 60–70 minutes. The participants' recruitment commenced after obtaining IRB approval and took place from February 1 to March 30, 2024.

### Data analysis

Upon completion of each interview, the recorded data underwent a verbatim transcription to Arabic and then was translated into English. For data organization and categorization, the English transcripts were entered into the NVivo 12 software (QSR International Pty Ltd, Doncaster, Victoria, Australia), which labels each paragraph with phrases representing specific ideas. The first author analyzed data using the interpretative phenomenological analysis framework [13]. The NVivo-generated categorized data was manually assigned as single phrases to groups of linked sub-phrases, which were then listed under a significant overarching idea. The research team held comprehensive discussions to identify themes predominantly zeroing in on extracted quotes from the interviews, which were then entered into the final thematic list.

**Table 1. Interview guide.**

| Questions |
| --- |
| **Perceptions of cesarean birth** |
| • What is your understanding of cesarean delivery? |
| • What do you believe are the benefits of cesarean delivery? |
| • What concerns or risks do you associate with cesarean delivery? |
| • How do you think cesarean delivery compares to vaginal birth? |
| **Decision-making factors** |
| • Can you describe the factors that influenced your choice of delivery method? |
| • How did your personal preferences play a role in your decision? |
| • What advice did you receive from your healthcare provider regarding delivery options? |
| • Were there cultural or societal factors that influenced your decision? If so, can you elaborate? |
| • Did you feel pressured by any external factors to choose a particular delivery method? |
| **Information and support** |
| • What kind of information and support did you receive about your delivery options? |
| • How helpful was the information provided by your healthcare provider? |
| • Did you seek information from other sources, such as family, friends, or online resources? |
| • How did your family or partner support you in making your decision? |
| • Were there any gaps in the information or support you received? If so, what were they? |
| **Post-delivery reflections** *(For participants who have undergone a cesarean section)* |
| • Reflecting on your cesarean delivery, how do you feel about the experience? |
| • Were you satisfied with the decision to have a cesarean section? Why or why not? |
| • How was your recovery process after the cesarean delivery? |
| • Did the experience meet your expectations? If not, how did it differ? |
| • How has the experience influenced your views on future deliveries? |

## Ethical consideration

Ethical clearance was obtained from the Institutional Review Boards of the Ministry of Health and the Royal Medical Services. The women were told that participation was voluntary and that their identities would not be disclosed. Informed consent was obtained from all participants before commencing the interviews.

## Trustworthiness of the study

The investigation adhered to the criteria of dependability, credibility, confirmability, and transferability to ensure its rigor [14]. The first author discussed the emerging themes and their interpretations with the research team, taking their feedback into account. During the study, several participants from the primipara focus groups 2 and 3 and the para 1 focus groups 2 and 4 expressed interest in reviewing the final list of themes derived from the interviews. To facilitate this, the thematic summaries were shared via WhatsApp with these participants, enabling them to provide feedback and ensure the accuracy of the findings with their experiences. This process ensured that participants' opinions, experiences, and needs were accurately represented.

## Results

Forty-one women were interviewed within seven focus groups. The women fell into the 19–49 age range. 17 women were primipara, and 24 were para 1, of whom 15 had delivered by cesarean section. 16 women were employed, and 25 were housewives. All the husbands were employed. 19 received ANC in public health facilities, 10 in private, and 12 mixed between public and private.

The thematic analysis of the Jordanian women's perceptions regarding CS is shown in Table 2.

Table 2.  Themes and sub-themes of the Jordanian women's perceptions regarding CS.

| Theme | Subtheme |
|---|---|
| The foundation of women's knowledge and cognitive structures | • Perceived meaning of CS<br>• Perceived benefits<br>• Perceived Concerns about CS<br>• Previous experiences |
| The parties influential in shaping women's perceptions | • Trust in the doctor<br>• Influence of the husband and gender equity<br>• Role of culture<br>• Maternal health condition |
| Transforming perception into action | • Decision on mode of delivery<br>• Recommendations for future mothers |

## The foundation of women's knowledge and cognitive structures

This theme explains how the woman's view of CS is formed and crystallized based on the knowledge she possesses and, ultimately, her feelings regarding her preferred mode of delivery. This theme consists of several sub-themes, as follows:

### Perceived meaning of CS

Some participants concurred that a CS does not represent the typical course of labor. Conversely, a 'normal' vaginal delivery (NVD) indicates an absence of complications and a smooth progression of labor. One participant explained:

*"NVD means a natural stage that the woman goes through at the end of her pregnancy. CS means there is a problem. Risk for the mother or the baby leads to having CS."* (Para 1, FG4)

Some women reported that a CS represented a bearable pain compared to the intense pain of NVD.

*"NVD means unbearable pain for me. Cesarean section is painful, but with strong painkillers, the pain is not comparable to the pain of labor."* (Para 1, FG 1)

### Perceived benefits of CS

Some participants said that they opted for CS because it meant that they would not be exposed to suffering and pain before seeing the baby. The pain after the CS could be tolerated as it would be alleviated by the presence of the newborn:

*"I prefer a CS because I will not suffer before I see my babies. I want to go to sleep and wake up to find them next to me. I feel that looking at them will ease my pain."* (Primipara, FG 3)

One participant added:

*"CS is easier and better for me. A CS is more comfortable, as the lady said. You sleep and wake up to find the baby next to you."* (Para 1, FG1)

Some women view CS as the preferable mode of birth because it does not entail waiting for the unknown. They appreciate the scheduled nature of the procedure, allowing for clear expectations regarding the date and mode of delivery. As one participant articulated:

*"I gave birth to CS. For sure, you are in pain after giving birth, but it is enough for you to enter the hospital without pain and go directly to the theatre; then, you and the baby will come out safely. You will suffer after giving birth, but God will make it easy."* (Para 1, FG2)

The factor of close and continuous monitoring of the mother during childbirth was a top priority among primipara women. They explained that one of the benefits of CS is that the doctor remains present from the beginning of the operation to its end, next to the woman giving birth. While in NVD, the doctor will only be present intermittently, coming at different times to examine her:

*"I intend to have a CS. The doctor is by your side. The anaesthesiologists are taking care of you and are paying attention. In NVD, it is normal for them to put you on the bed and forget about you. There is no care like that, I hear."* (Primipara, FG3).

Some women also felt that a woman who delivers by CS is given extra care because she has been through surgery and needs care until the wound heals.

*"A cesarean section means pampering and attention. You will stay asleep and comfortable... Oh, you will relax and be pampered for the first period."* (Primipara, FG2)

Some of the women confirmed their fears that NVD would alter the shape of the vulva and size of the vagina in a way that is unattractive, and this may affect the marital relationship, even leading to possible infidelity:

*"I know that the shape and size of the area will be changed after NVD. I heard the sight will be disgusting, and your husband may become disgusted to sleep with you, and he may cheat on you."* (Primipara, FG2)

This idea was reiterated and confirmed by other participants, who brought to light how their husbands directly requested a CS delivery, prompted by their concerns about the change in the shape of the vulva area and expansion of the vagina.

*"My husband went to the doctor and asked him for a CS. He discussed with the doctor the deformity of the area, and he couldn't bear to see this, and the doctor agreed to perform a cesarean."* (Para 1, FG4)

### Perceived concerns about CS

Some participants explained that the nature and duration of the pain were uppermost in their thoughts:

*"Honestly, both are painful, but I prefer NVD to CS because it will only hurt for hours, while CS will cause pain for weeks, and I will not be able to take care of my baby."* (Para 1, FG3)

One para 1 added that she was afraid of infection in the CS wound and was concerned about how to care for it:

*"I was afraid of infection from the operation and infection of the wound. I had no other fears."* (Para 1, FG3)

On the other hand, some women were aware of the dangers of CS and some complications that the mother may be exposed to, such as wound pain and possible infection, as well as long-term complications of adhesions. As for the fetus, he would be vulnerable to lack of oxygen:

*"A CS means adhesions, more bleeding, and a risk to the baby of low oxygen levels, requiring the baby to enter an incubator. CS means pain at the wound site, postpartum infections, lack of movement, and back needles."* (Para 1, FG2)

Some participants were excited about the CS but worried about their post-surgery condition and the resulting pain. They expressed concerns about being able to move about and care for the baby independently without help. They also thought about the future, their body shape, and the fear of sagging in the abdominal area, as one of the participants explained:

*"The thing I am most afraid of is coughing or sneezing after CS because they are very painful. Also, the first walk after the operation. Also, after I go home, when I wake up for the baby at night, I will sleep on my back; how will I get up? It will be a lot of suffering."* (Primipara, FG1)

## Previous experiences

Previous experiences greatly influenced the participants' perceptions regarding CS. It was clear that participants all knew about bad experiences, whether they were her own or those of a sister or friend. The trace of the experience remains engraved in the memory, often overshadowing many successful experiences. There are many women who were induced and went on to have an NVD, but what remains in the women's collective memory are the occasions when birth began with induction and ended, for mostly medical reasons, with a CS. In these cases, the women were exposed to the pain of normal delivery, added to the pain of a CS delivery:

*"When I was delivered, I lived through a week of terror, induction, and internal examination, opposite to what I expected. To be honest, I have been living in terror until now, from what I suffered."* (Para 1, FG 2)

Another participant added:

*"I preferred CS, to be honest; after what happened to my sister, I didn't want to give birth normally. My sister was admitted for labor, and a trainee doctor came and ruptured her membrane and the baby suffocated in her abdomen and died. I will tell them that I want a CS. I learned from the experience of my sister."* (Primipara, FG 1)

Some women narrated how, at the hospital, they witnessed other women who suffered during natural childbirth. They saw that there was negligence on the part of the medical staff and a lack of respect for the woman's privacy, so they felt discouraged about NVD and favoured CS:

*"From the front of the operating room, I could see a normal delivery room where a woman giving birth. She was bleeding, and no one paid attention to her. I tried to call the doctors and they said nothing was wrong. They did not come and see her. There was a lot of neglect and no care. The patient was also exposed in front of passers-by. This is unacceptable. In a CS, there is privacy, cleanliness, and care. Within a quarter of an hour, you are finished, and everything is fine, thank God."* (Para 1, FG1)

Some women told stories of relatives who were handled badly at birth and ongoingly experienced the consequences. A woman told us that when her husband was born, his shoulder was dislocated, and he had spent all his life unable to use his hand normally, so she preferred CS.

*"I am afraid of NVD. My husband had a normal birth, but when he was born, he had a dislocation in his shoulder. This thing affected him, and his hand is not normal until now. This thing scares me a lot. I don't want to give birth normally."* (Para 1, FG2).

### The parties influential in shaping women's perceptions

This theme discusses the parties who are influential in shaping women's opinions regarding CS delivery in both positive and negative ways.

### Trust in the doctor

It was evident that most of the participants trusted their doctor's opinion. The paternalistic concept was very clear during the participants' interviews. That is, the woman firmly believes that the doctor knows what is best for her condition and can make the appropriate decision:

*"He explained it to me exactly, from the beginning. He told me that I would have complications. So, I felt that he knew about my condition, and it was better to trust him."* (Primipara, FG2)

*"If the doctor had tried with me, I would have given birth normally. The decision is in the doctor's hands."* (Para 1, FG2)

### Influence of the husband and gender equity

Most women had discussed the preferred method of delivery with their husbands. Most husbands respected the doctor's opinion about the mode of delivery depending on the wife's health condition. Some husbands leave the decision to the wife, allowing her to do what is appropriate and comfortable for her:

*"He said the decision is in your hands and consult the doctor. No one feels the pain except you."* (Primipara, FG3)

*"My husband told me to do what I wanted. He said, "The important thing is that they save you."* (Primipara, FG1)

*"I used to talk to my husband a lot about the subject, and he encouraged me to deliver normally."* (Para 1, FG3)

### Role of culture

Cultural beliefs and norms surrounding childbirth heavily influence women's perceptions of CS. Jordanian culture sees vaginal birth as more natural or traditional. On the other hand, CS may be viewed as a safer option, entailing less suffering. Cultural attitudes towards pain, childbirth, and medical interventions can all shape how women perceive CS and influence their decision-making.

Our participants differed about whether the prevailing culture encourages NVD or CS, but ultimately, they agreed that the old generation and some of the new generation encourage NVD:

*"My mother and mother-in-law encourage NVD, while my friends encourage CS."* (Para 1, FG 1)

Many participants felt that most of the new generation of mothers encourages CS:

*"I am Primipara, but I worked as a maternity nurse. A week ago, I swear, 3 pregnant women asked for a CS. The doctor put them in for labor induction. They said that they didn't want to, so the doctor told them to sign a paper and performed a CS for them."* (Primipara, FG 1)

Several women expressed the belief that some doctors actively promote CS delivery:

*"I feel that women are not very enthusiastic about NVD, and doctors encourage women to have CS because of the financial factor involved in the matter. I feel that most new mothers have had cesarean births at present. My friend was pregnant with her fourth child. She had excellent conditions for a natural birth, but they referred her to have a CS without knowing the reason."* (Para 1, FG 1).

## Maternal health condition

A woman's health status and any potential complications during pregnancy can also impact her perception of CS. Pre-existing health conditions or pregnancy-related complications may increase the risk of a difficult NVD, thus encouraging the mother to consider a planned CS as a safer option for herself and her baby.

*"I had no dilatation and no contractions. The cervix was closed. At the end, the baby's pulse began to decrease because the fluids were very light, because of this factor. The only reason I chose CS was because there was a threat to the life of the fetus."* (Para 1, FG 4)

*"I conceived through IVF. What made me give birth by CS was that I gave birth to her in the eighth month. My water broke, and I had to go to the hospital. There, they started telling me to wait, but I told them that I was not ready to take the risk, not even 1%. I did not agree with the NVD because I told them that I had waited all these years to see the baby. That's why I requested CS."* (Para 1, FG2)

## Transforming perception into action

Perceptions of CS delivery, therefore, are built on both preconceived notions (stemming from personal or familial experience) and objective open-mindedness (allowing physician intake). We understand how knowledge was formed among the participants and the factors that led to the formation of women's perception of CS, thus the main theme remains on transforming these perceptions into reality.

## Decision on mode of delivery

Some women asserted their autonomy in the decision-making process, and their husbands' sole concern was the safety of mother and baby. Their words echoed the sentiment of empowerment and safety, highlighting their trust in their decision and the support of their partner.

*"The decision to choose the method of delivery is up to me. My husband tells me to do what I feel comfortable about. The important thing is that you and the baby are safe."* (Primipara, FG 2)

Similarly, some women showed a sense of independence and self-assurance in expressing their desire for a C-section, committed to their own beliefs and unaffected by external pressures or opinions:

*"I want to give birth by CS. I don't feel that those around me are influencing me. I don't listen to their opinions. I told you; I only do what I am convinced of."* (Primipara, FG 3)

Other women see the doctor as pivotal in the decision-making process, and they are prepared to change their decisions based on medical advice.

*"The doctor is first in line. He knows my condition and that of the fetus. I mean, I like NVD. If he tells me that a CS is better, I will respond."* (Primipara, FG 1).

## Recommendations for future mothers

When asked what recommendations they would give to women about giving birth, most of the participants were aware of the impact of their advice but chose not to express their opinion. They emphasized the importance of informed decision-making guided by medical expertise based on each mother's particular circumstances. They avoided imposing their own experiences on other expectant mothers, honoring their autonomy to make their own decisions.

*"I will not express my opinion. She may take my words seriously. I will tell her, "See what suits you and do it. See what the doctor tells you."* (Primipara, FG2)

*"According to her doctor and his advice. According to her condition and that of her fetus. I won't tell her about my experience."* (Para 1, FG3)

## Discussion

This study's findings show that cultural, social, and individual factors influence women's perceptions of CS. Some women perceived CS as a necessary medical intervention to ensure the safety of both mother and baby, viewing it as a modern advancement that offers a controlled and planned delivery while potentially reducing the risks associated with NVD, this finding was incongruent with the results of Longo et al. [12]. However, this perception is not always grounded in medical necessity but may stem from fear, misinformation, or cultural beliefs. It is critical for healthcare providers to engage in mutual, open communication with expectant mothers, ensuring that decisions for CS are based on accurate medical indications rather than convenience or societal perceptions. In cultures where NVD is traditionally preferred, CS often carries negative associations, representing intervention and a deviation from the normal process of childbirth [13]. Healthcare providers must acknowledge and address these diverse perspectives to provide tailored support, education, and informed decision-making to expectant mothers.

Most of our participants perceive CS as a safer and less painful alternative to NVD. A study conducted in Pakistan reported that women chose CS because they feared the pain associated with NVD, and they preferred to schedule the delivery [15]. In our study, women self-reported that their obstetricians and their recommendations were a major influence.

Despite some women relying on specific knowledge areas and personal or family experiences to choose their mode of delivery, a power hierarchy exists between them and obstetricians. This often results in medical professionals making the final decision. Women and their

husbands frequently expressed trust in the doctors, placing the final decision in their hands. As observed in [16], although pregnant women expressed a preference for NVD, their wishes are often overlooked due to the dominance of CS preference in obstetrics, which can stem from institutional protocols, practitioner convenience, or liability concerns. This underscores the need for healthcare providers to prioritize patient-centered care, ensuring that women's preferences for NVD are respected and supported whenever medically appropriate. Moreover, the narrative literature review by Sys et al. (2021) asserted that communication with medical professionals is the key to making an informed decision regarding the mode of delivery. These results can help obstetricians identify and acknowledge their role as crucial members in the decision-making process for CS deliveries within their institutions. They can also form the basis for the development of an intervention that targets women and their husbands during the antenatal period, raising awareness of the benefits of NVD and the risks of CS. Obstetricians and other maternity healthcare providers have a duty to try to change women's negative perceptions of natural childbirth, so they don't opt for medically unjustified CS deliveries.

Krychman (2016) opined that vaginal laxity can decrease sexual satisfaction, which can affect a woman's sense of sexual self-esteem and her relationship with her sexual partner. The women in the current study revealed that the fear of vaginal laxity following NVD is a significant concern for them and their husbands, causing them to opt for CS over NVD. They worry about a loss of vaginal tightness, which they believe could negatively affect sexual satisfaction for both themselves and their husbands. This belief can lead women to reject natural childbirth due to the potential consequences of vaginal delivery, ignoring the higher medical risks and longer recovery associated with CS. It also underscores the mix of psychological and social factors that influence childbirth decisions. A holistic approach is needed to address childbirth's full range of medical, emotional, and relational dimensions. Comprehensive education is called for so that healthcare providers can support women in making informed decisions while reducing the health risks of unnecessary medical procedures.

### Limitations of the study

It should be noted that the perceptions of our participants are not representative of all primipara and para 1 Jordanian women, whose experiences may differ. The analysis presented in this study emerged from the experiences of the women in our sample, which included varying opinions regarding cesarean sections (CS), with some in favor and others against. We found that women generally followed an individualized decision-making process regarding CS, shaped by their knowledge, personal context, and several influencing factors while placing significant trust in their doctors' recommendations and engaging in a collaborative decision-making process. However, specific contexts and cultural settings may limit the generalizability of these findings. This limitation, inherent to qualitative research, is due to its focus on in-depth exploration of experiences within a specific group rather than statistical representation. As such, the viewpoints of this subset of Jordanian primiparous and para 1 women may not be universally applicable. Future research could benefit from broader sampling and mixed-method approaches to enhance the generalizability and applicability of these findings

### Conclusion

In conclusion, women hold a complex blend of ideas and perceptions when it comes to modes of birth. Although they have their own perceptions regarding the meaning, benefits, and disadvantages of delivery by Cesarean section, the major influence on their decision-making was ultimately found to be the advice they received from their doctor. A full understanding of

the diverse perspectives and preferences of expectant women when it comes to the mode of delivery can potentially inform patient-centered care approaches where women are encouraged to make informed decisions while receiving psychological support and holistic antenatal care. Furthermore, the findings of this study highlight the importance of integrating women's perspectives into policy development to enhance patient-centered maternity care. Policies should prioritize shared decision-making between women and healthcare providers, emphasizing the importance of informed consent and providing clear, evidence-based information about cesarean delivery and its implications. Additionally, the study underscores the need for policies that support education and counseling programs to address non-medical requests for cesarean sections, ensuring that maternal and neonatal health outcomes are optimized. These findings can guide the development of maternity care policies in Jordan and similar contexts, promoting better alignment between women's preferences and evidence-based clinical practices

## Acknowledgments

This manuscript is made possible by the support of the American people through the United States Agency for International Development (USAID) in partnership with the Ministry of Health. The contents/findings of this manuscript are the sole responsibility of University Research Co., LLC (URC) and do not necessarily reflect the views of USAID or the United States Government.

## Author contributions

**Conceptualization:** Zaid Al-Hamdan, Nagham Abu Shaqra, Majida Jallad.

**Data curation:** Majida Jallad, Heba AboShindi.

**Formal analysis:** Hala Bawadi, Hasan Rawashdeh.

**Methodology:** Hala Bawadi, Amer Gharaibeh, Asma Basha.

**Project administration:** Zaid Al-Hamdan, Sawsan Majali.

**Resources:** Nagham Abu Shaqra, Majida Jallad, Sawsan Majali, Heba AboShindi.

**Supervision:** Nagham Abu Shaqra, Sawsan Majali, Mahmoud Taym.

**Validation:** Amer Gharaibeh, Hasan Rawashdeh, Heba AboShindi, Mahmoud Taym.

**Writing – original draft:** Hala Bawadi.

**Writing – review & editing:** Hasan Rawashdeh, Asma Basha.

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
