## [Decision Letter · Decision Letter 0]

2 Oct 2024

PONE-D-24-38372An Interpretative Phenomenological Study about Maternal Perceptions of Cesarean BirthPLOS ONE

Dear Dr. Al-Hamdan,

Thank you for submitting your manuscript to PLOS ONE. After careful consideration, we feel that it has merit but does not fully meet PLOS ONE’s publication criteria as it currently stands. Therefore, we invite you to submit a revised version of the manuscript that addresses the points raised during the review process.

Please submit your revised manuscript by Nov 16 2024 11:59PM.  If you will need more time than this to complete your revisions, please reply to this message or contact the journal office at plosone@plos.org . Please include the following items when submitting your revised manuscript:

We look forward to receiving your revised manuscript.

Kind regards,

Mergan Naidoo, PhD

Academic Editor

PLOS ONE

Journal Requirements:

Reviewers' comments:

Reviewer's Responses to Questions

**Comments to the Author**

1. Is the manuscript technically sound, and do the data support the conclusions?

Reviewer #1: Yes

Reviewer #2: Yes

Reviewer #3: Yes

2. Has the statistical analysis been performed appropriately and rigorously? 

Reviewer #1: Yes

Reviewer #2: N/A

Reviewer #3: Yes

3. Have the authors made all data underlying the findings in their manuscript fully available?

Reviewer #1: No

Reviewer #2: No

Reviewer #3: Yes

4. Is the manuscript presented in an intelligible fashion and written in standard English?

Reviewer #1: Yes

Reviewer #2: Yes

Reviewer #3: Yes

5. Review Comments to the Author

Reviewer #1: This article is a standard piece of qualitative work. it is of interest to the reader who works in the obstetrical arena. It does not have mind blowing results, but it answers a local question. The authors do acknowledge that the findings are not transferable. it remains of interest that in their sample many women found that although asked they did not make the final decision and a C/S.

Answered no to Q3 as there must be many more data available that the authors do not mention.

Reviewer #2: An interesting article. I have made a few comments in the manuscript and in the attached notes.

It would be useful to give an overview of maternal care in Jordan. Who provided Antenatal care, what is covered in the educational material, who decides about delivery methods? is this a joint decision? does the women have any autonomy or is it mainly based on medical indications in the government hospitals?

The abstract talks about focus groups and semi structured interviews. However the methods section only talks about focus group discussions.

The methods talk about maximum variability but this is not reflected in the results. It would have been important to know who in the P1 group had had a CS delivery

It is not clear how participants were actually selected or how saturation of themes was reached

Primips don't have much experience with either NVD or CS - it would be important to know who shapes their perceptions - this is not clear from the data presented

I like the theme heading to come from the data

The discussion is not structured in the same order as the results

Aims speak to policy development. I could not see anything in the conclusions about policy development

Reviewer #3: Thank you for having me review your work. It makes an important contribution in understanding how women make decisions around their preferred mode of delivery. The data gives granular insight into the pSome repsonses from participants bring light to providers not necessarlly exploring the woman's reasoning for deciding on a mode of delivery and waht meaning it has for her. The contribution indeed makes inroads in aspects of patient-centred, respectful care that need further exploartion.

6. PLOS authors have the option to publish the peer review history of their article (what does this mean? ). If published, this will include your full peer review and any attached files.

**Do you want your identity to be public for this peer review?** For information about this choice, including consent withdrawal, please see our Privacy Policy .

Reviewer #1: No

Reviewer #2: **Yes: ** Andrew Ross

Reviewer #3: **Yes: ** Jason K. Marcus

---

## [Author Response · Author response to Decision Letter 0]

24 Dec 2024

Replay to reviewers comments

Comments Replay Page number

Methods – line 97 indicates that only focus group discussion were done "Yes, only seven focus groups were conducted. All instances of 'in-depth interview' in the text were changed to 'focus group discussions (FGDs).'" Page 2+ 5+ 6

Results: I think that it is better if the theme headings come from the content of the interviews – I am not sure that women would be using the words: cognitive structures and transforming perceptions into practice Thank you for highlighting this aspect. In interpretive phenomenological analysis (IPA), the researcher’s role includes making sense of participants' experiences through their interpretations, which often involve conceptualizing themes beyond participants' language. The themes reflect the researcher's analytical depth and insight into the phenomenon rather than staying strictly at the participants' level of language and knowledge. This study's analysis was based on our interpretations and intended to provide deeper insights into the participants’ experiences rather than mirroring their exact language. The thematic headings aim to capture the essence of the experiences from participants' narratives. Your feedback is highly appreciated and respected.

Introduction

An overview of the health care services in Jordan would be helpful. How much say do women have in whether or not they have a CS? If they would prefer a CS is this taken into consideration or are most CS done for medical indications? Who provides antenatal care? What is covered during antenatal care? Thank you for these valuable suggestions. We agree that including an overview of the healthcare services in Jordan, specifically regarding maternal and childbirth care, would enrich the context of the study. We added a section detailing the healthcare structure, accessibility, and the provision of maternal services. We will also address the degree of autonomy women have in deciding to undergo a cesarean section. Thank you again for your helpful comments. Incorporating the context of maternal health care services will strengthen the contextual background and enhance the study’s relevance and clarity. The following paragraphs are added to the manuscript:

In Jordan, maternity care encompasses both public and private healthcare services, supporting women through pregnancy, childbirth, and the postpartum period. While many women utilize public health facilities due to their accessibility and affordability, a significant number choose private healthcare for its perceived higher quality and personalized care, despite the associated higher costs. Public hospitals often remain the primary choice for complex pregnancies or labor due to their comprehensive services (Khader et al., 2018; Alyahya et al., 2019).

Maternity care services are provided by obstetricians, nurses, and midwives across both healthcare sectors. Notably, 96% of women prefer to receive antenatal care from a doctor, with only 3% selecting for midwives or nurses, indicating a strong preference for physician-led services. Antenatal visits typically include essential check-ups, screenings, nutritional advice, and health education. The experience can vary based on the chosen healthcare setting: private facilities often provide longer consultations, which many women find more reassuring, while public services are favored for their affordability and availability (Alyahya et al., 2019).

Women’s perceptions and decisions about CS are shaped by personal beliefs, cultural influences, and healthcare setting. Some women view CS as a necessary medical intervention to ensure the safety of both mother and baby, perceiving it as a modern advancement that offers controlled and planned delivery while potentially reducing the risks associated with normal vaginal delivery (NVD). However, this perception is often influenced by factors such as trust in doctors, advice from healthcare providers, and familial or cultural expectations, particularly in private hospitals where personalized care is emphasized (Fram et al., 2023). Page 3+ 4

Why were primips chosen? They may have perceptions, but they don’t have any experience of giving birth either vaginally or via CS Thank you for your valuable comment. One of the aims of the study was to understand women's perspectives on the cesarean section. To achieve this, we included both primiparous women and those who had given birth once (para 1). Primiparous women were specifically included because their perceptions are influenced by societal, cultural, and healthcare-related factors, which shape their opinions and expectations about cesarean birth, even without personal birthing experience. Interviews conducted during pregnancy provided an opportunity to explore their perspectives and decision-making processes before they experienced childbirth.

Additionally, the inclusion of women who had given birth once allowed us to examine how these initial perceptions may change after their first birth experience, regardless of the mode of delivery. This approach enabled a more comprehensive understanding of maternal perspectives on cesarean sections, both before and after giving birth. We believe this diversity in the sample enriched the study by capturing the full picture of experiences and expectations related to cesarean delivery.

I am also not sure how women’s perspectives – particularly if they are wanting a CS for non-medical reasons, promote optimal maternal and neonatal outcomes (line 71 – 72). I do understand that there is a need for mutual dialogue and a shared understanding of the reason for CS. Thank you for your insightful comment. We appreciate your comment regarding how women's perspectives, particularly in cases where they desire a cesarean section for non-medical reasons, align with promoting optimal maternal and neonatal outcomes.

Our intent in this section was to highlight the importance of understanding maternal perspectives and encouraging a discussion between women and healthcare providers. We recognize that decisions for cesarean sections driven by non-medical reasons may not always align with optimal clinical outcomes. However, we believe that engaging in mutual discussion and establishing a shared understanding of the reasons for cesarean delivery is essential in ensuring that women are well-informed about potential risks and benefits. This collaborative approach supports decision-making that prioritizes both maternal preferences and evidence-based practices for maternal and neonatal health.

To clarify, in our recommendations, we mentioned the importance of informed decision-making and the role of healthcare providers in guiding women toward choices that support optimal outcomes while respecting their preferences. Thank you for bringing this to our attention.

The purpose of the study was to provide insight for clinical practice, policy development and future practice (line 81 – 82). In the conclusions (line 390 – 396) I cannot find reference to how the finding contribute to policy development We appreciate your feedback on ensuring that the conclusions adequately address how the findings contribute to policy development. We agree that linking the findings to policy development is essential to fulfill the stated purpose of the study.

To address this, we will revise the conclusion section to clearly outline how the study findings can inform policy development in maternity care. Specifically, we will highlight the importance of incorporating women's perspectives into policymaking to create patient-centered approaches, improve informed consent processes, and address non-medical cesarean section requests through education and counseling policies.

This is the section we added to the conclusion:

Furthermore, the findings of this study highlight the importance of integrating women's perspectives into policy development to enhance patient-centered maternity care. Policies should prioritize shared decision-making between women and healthcare providers, emphasizing the importance of informed consent and providing clear, evidence-based information about cesarean delivery and its implications. Additionally, the study underscores the need for policies that support education and counseling programs to address non-medical requests for cesarean sections, ensuring that maternal and neonatal health outcomes are optimized. These findings can guide the development of maternity care policies in Jordan and similar contexts, promoting better alignment between women's preferences and evidence-based clinical practices Page 22

Setting and participants

Lines 57 – 60 suggest that high rates of CS are found in University hospitals (40,4%) Private hospitals (39,1%), Military hospitals (36,1%) and government hospitals (27,4%) but participants were chosen from 2 public hospitals and 2 health centres (do they even do CS at health centres?). Why were these centres chosen and not those with high rates of CS? Thank you for your comment. We appreciate the opportunity to clarify the rationale behind our selection of settings and participants for this study.

While high rates of cesarean sections were noted in university, private, and military hospitals, we aimed to gather a representative sample of perspectives from women across various healthcare settings in Jordan. To achieve this, we included participants from one military hospital and one teaching hospital, both of which perform cesarean sections, and complemented this with participants from two health centers. Although health centers do not provide cesarean sections, they offer postpartum care and newborn vaccinations, making them an important point of contact for mothers who have recently given birth. This approach allowed us to reach women who delivered in various types of hospitals, including private and government hospitals, thereby capturing a broader range of views and experiences regarding cesarean sections.

We believe this sampling technique (maximum variation sampling) enabled us to gather diverse insights into maternal perceptions of cesarean birth across Jordan, reflecting the different healthcare contexts where women receive maternity care.

Line 92 – 93 - 3 +4 = 7 Focus groups while the abstract suggests that in depth semi structured interviews were also done (line 39 -40). Please clarify Thank you for pointing out this inconsistency. We appreciate your careful review and feedback on this matter.

To clarify, we conducted 7 focus group interviews, all of which were guided by a semi-structured format. The phrasing "in-depth semi-structured" was mistakenly included in the manuscript, which may have led to confusion. To address this, we have deleted all references to "in-depth semi-structured" from the manuscript and ensured that the description of the methodology is consistent throughout.

We hope this revision resolves the issue and provides clarity regarding the methodology used in this study. Thank you for bringing this to our attention. Page 2+ 5+ 6

Line 95 talks about maximum variation – It is not clear what factors were considered in selecting women with maximum variation. Age range, and employment are reported in the results but not education or family income Thank you for your valuable comment. We appreciate the opportunity to clarify the factors considered in achieving maximum variation in participant selection.

In selecting participants, we aimed to capture diverse experiences and perspectives regarding cesarean birth. The primary factors considered for maximum variation included age, parity (primipara vs. para 1), employment status (employed vs. housewives), and the type of antenatal care (public, private, or mixed). While we did not specifically include education level or family income, the diversity in the settings from which participants were recruited from public hospitals, military hospitals, teaching hospitals, private hospitals, and health centers—likely contributed to variability in socioeconomic backgrounds.

Line 97 – it is not clear how theme saturation influences the sample size. Where there interviews, then analysis and theme identification, followed by more interviews, analysis and theme identification? Thank you for your comment, regarding the concept of 'theme saturation' and its influence on sample size. We appreciate the opportunity to clarify our methodology and address the terminology used.

We acknowledge that the term 'theme saturation' may not be standard terminology in qualitative research. The more widely accepted term is 'data saturation,' which refers to the point in data collection when no new information or themes emerge, indicating that the data is sufficiently rich and comprehensive. We will revise the manuscript to replace 'theme saturation' with 'data saturation' to align with established qualitative research terminology.

Thank you for bringing this to our attention, and we will make the necessary revisions to enhance the clarity and accuracy of our manuscript.

Page 6

Line 100 – please clarify if in depth interview were done. It would appear that 7 focus groups X 5 – 7 members = the 41 participants Thank you for pointing out this inconsistency. We appreciate your careful review and feedback on this matter.

To clarify, we conducted 7 focus group interviews, all of which were guided by a semi-structured format. The phrasing "in-depth semi-structured" was mistakenly included in the manuscript, which may have led to confusion. To address this, we have deleted all references to "in-depth semi-structured" from the manuscript and ensured that the description of the methodology is consistent throughout.

We hope this revision resolves the issue and provides clarity regarding the methodology used in this study. Thank you for bringing this to our attention.

Page 2+ 5+ 6

Line 102 – it is not clear how the questions were developed and what questions were asked. These should be included as an appendix Thank you for your comment. The research team developed a semi-structured interview guide to explore maternal perceptions of cesarean birth. This guide was informed by a comprehensive review of existing literature on maternal health and cesarean delivery, as well as consultations with experts in obstetrics and qualitative research methodologies. To provide a comprehensive understanding of our methodology, we will include the complete interview guide as an appendix in the revised manuscript. This addition will offer readers detailed insight into the specific questions posed during the interviews. We hope that this clarification addresses your concerns, and we appreciate your constructive feedback, which has contributed to improving the clarity and rigor of our study. Page 6

Line 105 – when did the focus group discussions take place? The dates of the interviews has been added: “Participant recruitment commenced after obtaining IRB approval and took place from February 1 to March 30, 2024.” Page 6

Line 107 – it is not clear how the selection of participants was done. What was the process of selecting participants Thank you for your insightful comment regarding the participant selection process. To ensure a diverse and representative sample, we employed a purposive sampling strategy, focusing on maximum variation to capture a wide range of maternal perspectives on cesarean birth. Women who met the inclusion criteria were invited to participate in the study.

Line 130 – who were the WhatsApp sent to? Thank you for your insightful question regarding the recipients of the WhatsApp messages mentioned in line 130. We appreciate the opportunity to clarify this aspect of our methodology.

During the study, several participants from the primipara focus groups 2 and 3, as well as from the para 1 focus groups 2 and 4, expressed interest in reviewing the final list of themes derived from the interviews. To facilitate this, we utilized WhatsApp to share the thematic summaries with these participants, enabling them to provide feedback and ensure the accuracy of the findings with their experiences. We will revise the manuscript to include this clarification.

Thank you for bringing this to our attention. This is the revised text:

The first author discussed the emerging themes and their interpretations with the research team, taking their feedback int

---

## [Editor Report · Decision Letter 1]

17 Jan 2025

An Interpretative Phenomenological Study about Maternal Perceptions of Cesarean Birth

PONE-D-24-38372R1

Dear Dr. Zaid Al-Hamdan

We’re pleased to inform you that your manuscript has been judged scientifically suitable for publication and will be formally accepted for publication once it meets all outstanding technical requirements.

Kind regards,

Mergan Naidoo, PhD

Academic Editor

PLOS ONE
---

## [Editor Report · Acceptance letter]

PONE-D-24-38372R1

PLOS ONE

Dear Dr. Al-Hamdan,

I'm pleased to inform you that your manuscript has been deemed suitable for publication in PLOS ONE. Congratulations! Your manuscript is now being handed over to our production team.

Kind regards,

on behalf of

Professor Mergan Naidoo

Academic Editor

PLOS ONE